# How Does Orthodontic Mini-Implant Thread Minidesign Influence the Stability?—Systematic Review with Meta-Analysis

**DOI:** 10.3390/jcm11185304

**Published:** 2022-09-09

**Authors:** Maciej Jedliński, Joanna Janiszewska-Olszowska, Marta Mazur, Katarzyna Grocholewicz, Pedro Suárez Suquía, David Suárez Quintanilla

**Affiliations:** 1Department of Interdisciplinary Dentistry, Pomeranian Medical University in Szczecin, 70-111 Szczecin, Poland; 2Department of Dental and Maxillofacial Sciences, Sapienza University of Rome, 00161 Rome, Italy; 3Department of Surgery and Medical-Surgical Specialities, Faculty of Medicine and Dentistry, University of Santiago de Compostela, 15705 Santiago de Compostela, Spain

**Keywords:** orthodontic mini-implant, temporary anchorage device, thread, design, stability, anchorage

## Abstract

Background: Clinical guidelines are lacking for the use of orthodontic mini-implants (OMIs) in terms of scientific evidence referring to the choice of proper mini-design. Thus, the present study aimed to investigate to what extent orthodontic mini-implant thread design influences its stability. Methods: Search was conducted in five search engines on 10 May. Quality assessment was performed using study type specific scales. Whenever possible, meta-analysis was performed. Results: The search strategy identified 118 potential articles. Twenty papers were subjected to qualitative analysis and data from 8 papers—to meta-analysis. Studies included were characterized by high or medium quality. Four studies were considered as low quality. No clinical studies considering the number of threads, threads depth, or TSF have been found in the literature. Conclusions: Minidesign of OMIs seems to influence their stability in the bone. Thread pitch seems to be of special importance for OMIs retention—the more dense thread—the better stability. Thread depth seems to be of low importance for OMIs stability. There is no clear scientific evidence for optimal thread shape factor. Studies present in the literature vary greatly in study design and results reporting. Research received no external funding. Study protocol number in PROSPERO database: CRD42022340970.

## 1. Introduction

A few decades ago, introducing skeletal anchorage revolutionized orthodontic treatment dogma and significantly broadened therapeutic possibilities [1]. With the use of orthodontic mini-implants (OMIs), it is possible to achieve direct or indirect anchorage for a specific tooth movement as well as skeletal anchorage for facemasks or hybrid maxillary expansion appliances [2,3]. Thus, the possibility of orthodontic treatment is increased, less patient’s compliance may be required to obtain a treatment goal and the need for orthognathic surgery may be reduced [4]. The increasing popularity of orthodontic OMIs is due to a simplicity of the surgical technique, easy acceptance of OMIs by patients, and low cost regarding the effect achieved [5]. Innovative materials and technologies to improve implants primary stability are an intense research topic in dentistry [6]. The use of skeletal anchorage has been the subject of above 1500 scientific papers published in the recent two decades [7], and manufacturers provide more and more innovative solutions. As a result, multiple systematic reviews regarding the influence of OMIs geometry on treatment success rate can be found, generalizing and assessing multiple factors [7,8,9,10,11] and thematically specific [12,13]. Clinical guidelines can be found in the literature referring to the influence of macrodesign of OMIs on success rate.

The factors potentially influencing success rate are:

Characteristics of the screw geometry: such as screw diameter (≤1.3 mm) and length (≥8 mm) proved important for the success rate [7,8,9]. However, the design of thread taper (microdesign) has not been widely assessed so far. Thus, clinical guidelines are lacking for the use of OMIs in terms of scientific evidence referring to the clinical choice of proper microdesign characteristics of OMIs. Thus, the present study aimed to investigate to what extent orthodontic mini-implant thread design influences the stability of OMIs.

## 2. Materials and Methods

### 2.1. Search Strategy

This systematic review was conducted in accordance with the PRISMA statement [14], reporting guidelines [15,16] (Appendix A), and the guidelines from the Cochrane Handbook for Systematic Reviews of Interventions [17]. On 6 May 2022, a series of pre-searches of the following databases was performed: PubMed, PMC, Scopus, Web of Science, Embase. Then, the study protocol was registered in the PROSPERO database (Ref. No CRD42022340970) on 10 May 2022. Subsequently, the final search was proceeded on 12 May 2022 using the following search engines: PubMed, PubMed Central, Web of Science, Scopus, Embase, with the following keywords: (“mini-implant” OR “miniscrew” OR “TAD” OR “temporary anchorage device” OR “skeletal anchorage”) AND (“orthodontics” OR “malocclusion” OR “Tooth Movement Techniques”) AND (“thread” OR “thread pitch” OR “thread depth” OR “thread shape” OR “thread design”) AND (“success rate” OR “success” OR “successful” OR “survival rate” OR “failure risk” OR “treatment success” OR “stability”). The exact search string for every search engine applied is described on PRISMA 2020 flow diagram (Figure 1). In accordance with PICO(S) [18], the framework of the present systematic review is Population: orthodontic mini-screws, Intervention: skeletal anchorage insertion; Comparison: orthodontic mini-screw stability of with different thread mini-design; Outcomes: pull-out strength. Study design: prospective clinical and animal studies, in-vitro studies, finite element analysis. The PICO(S) question was the following: “Does orthodontic mini-implants thread design influence their stability within the bone?”.

### 2.2. Eligibility Criteria

For the present systematic review, the following inclusion criteria were applied:

**Type of study:** prospective clinical and animal studies, in-vitro studies, finite element analysis. **Results of the study:** pull-out strength, removal torque **Object of the study:** evaluation of the influence of orthodontic mini-implant design on its stability

**Subject of the study:** orthodontic mini-implants

The following exclusion criteria were as follows:

Studies not referring to the design of orthodontic mini-implants, in-vivo retrospective studies, ex-vivo studies not using finite element analysis, case reports, reviews, authors’ opinions, conference reports, studies lacking effective statistical analysis, studies considering the type of material used, studies evaluating the effectiveness of specific orthodontic movement with the use of skeletal anchorage, studies evaluating the influence of biological factors on skeletal anchorage effectiveness. No language restriction was applied.

### 2.3. Data Extraction

After retrieving the results from search engines to create a database, duplicates were removed. Then, titles and abstracts were analyzed by two authors independently (MJ and MM), following the inclusion criteria. Subsequently, the full text of each selected article was analyzed to verify, whether it was suitable for inclusion and exclusion criteria. Whenever disagreement occurred, it was resolved by discussion with the third author (DSQ) by creating a working spreadsheet in order to verify the accordance with Cochrane Collaboration guidelines [17]. The Cohen’s K coefficient for the agreement between the authors indicates a high agreement between the authors and was equal to 0.91, due to the gross number of articles on this topic. Authorship, year of publication, type of each eligible study, and its relevance regarding the orthodontics mini-implant design were extracted by one author (DSQ) and examined by another author (MJ).

### 2.4. Quality Assessment

According to the PRISMA statements evaluation of methodological quality must be performed in order to properly assess the strength of evidence provided by the included studies, as methodological flaws can result in biases [14].

Due to a wide range of types of studies that were finally included in this review (animal studies, finite element analysis, in-vitro studies) the authors decided to use three types of specific quality assessment tools—SYRCLE for animal studies [19], Methodological Quality Assessment of Single-Subject Finite Element Analysis Used in Computational Orthopedics for finite element analysis (MQSSFE) [20] and QUIN for in-vitro studies [21]. While assessing the studies according to the SYRACLE assessment tool, 10 different types of bias were evaluated that possibly could have occurred. Two authors independently assessed the risk of bias by scoring “+” if there was no risk of bias in the assessed category, “−“ if the possibility of bias occurred, and “?” when it was impossible to assess wherever it occurred or not. MQSSFE consists of 37 questions and is evaluated independently by two researchers. If there was no risk of bias, “YES” was entered, and “No” if there was a risk of bias. If the researchers disagree on a point in the checklist, a half point is issued for a given checklist question. In the case of the QUIN assessment tool, two independent authors (MJ and MM) evaluated independently each of the 12 criteria as adequately specified = 2 points, inadequately specified = 1 point, not specified = 0 points, and not applicable = exclude criteria from the calculation. Then, the scores were summarized to obtain a total score for a particular in vitro study. The scores thus obtained were used to grade the in vitro study as high, medium, or low risk (>70% = low risk of bias, 50% to 70% = medium risk of bias, and <50% = high risk of bias).

### 2.5. Meta-Analysis

Meta-analysis was performed with the R statistical software (R Foundation, Vienna, Austria), ver.4.1.2 [22] using a random-effect model via metafor R package [23], with Mean Differences (MD) and 95% confidence intervals (95% CI) being calculated as effect estimates. Heterogeneity was assessed quantitatively using I2-statistics and Cochran’s Q [24]. The results were considered statistically significant at *p* < 0.05. Publication bias was estimated using a funnel plot.

## 3. Results

### 3.1. Results of the Search

The search strategy identified 118 potential articles: 20 from PubMed, 45 from PubMed Central, 23 from Scopus, 18 from Web of Science, and 12 from Embase. At the beginning of the analysis, 22 duplicates were removed, and 96 titles and abstracts were analyzed. Subsequently, 61 papers were excluded because they did not meet the inclusion criteria (completely different subject; studies regarding other factors affecting MI stability, systematic reviews). Of the remaining 34 papers, only 1 could not be retrieved. Fourteen studies had to be excluded, because they were not relevant to the subject of the study (discussing a different topic, not including taper design into the evaluated factors, retrospective analysis, or in one study—lack of an effective statistical analysis). Thus, finally, 20 papers were subjected to qualitative analysis, and data from 6 papers were subjected to meta-analysis. Two in-vitro studies did not provide exact resulting values, giving only the relationships between the tested parameters. The latter five studies did not provide sufficient data. The whole procedure is described in Prisma 2020 Flow Diagram (Figure 1. Flow diagram) The main characteristics of each included study are presented in Table 1.

### 3.2. Quality Assessment

The results of the assessment are presented in Table 2, Table 3 and Table 4.

From the quality analysis performed, it can be concluded that 1 animal study [25] and 2 finite element analyses are at high risk of bias [27,29], and most of the in-vitro studies are at medium bias risk. Two finite element analyses and two in-vitro studies should be considered at low risk [28,30,35,39].

### 3.3. Meta-Analysis

Even if studies included in the review may seem possible to be included in meta-analysis they had to be excluded since they presented only correlations between the examined factors, not specific values. [35,41] In the other study there was a different research material used (human cadaver heads). [42] This is a significant loss for the study, because the studies mentioned were designed similarly, and the number of OMIs tested was significant. Each of the included studies was based on the same polyurethane foam block in case of artificial bone and in case of animal bone—on similar species of animals in similar conditions on similar insertion depth. The data used to perform meta-analysis are summarized in Table 5. N1/N2 are numbers of OMIs in the left/right part of the table. Negative values of MD mean smaller dimensions of a given diameter in OMIs in the left part of the data table.

#### 3.3.1. Meta-Analysis of In-Vitro Studies of Peak Load for Pull-Out Strength Regarding Thread Pitch Dimension

(A)artificial bone model

There is small insignificant (*p* = 0.053) negative effect size. Study results are inconsistent—heterogeneity is significant (*p* < 0.001), and almost 98% of the variability comes from heterogeneity (Figure 2). The funnel plot confirms high heterogeneity, asymmetry suggests some publication bias (Figure 3).

(B)animal bone model

There is small significant (*p* = 0.005) negative effect size. Study results are consistent—heterogeneity is insignificant (*p* = 0.157), and only about 40% of the variability comes from heterogeneity (Figure 4). The funnel plot does not reveal publication bias (Figure 5).

#### 3.3.2. Meta-Analysis of In-Vitro Studies of Peak Load for Pull-Out Strength Regarding Thread Depth

(A)artificial bone model

There is small insignificant (*p* = 0.117) negative effect size. Study results are inconsistent—heterogeneity is significant (*p* < 0.001), and more than 98% of the variability comes from heterogeneity (Figure 6). The funnel plot confirms high heterogeneity, asymmetry suggests some publication bias (Figure 7).

(B)animal bone model

There is small insignificant (*p* = 0.243) negative effect size. Study results are inconsistent—heterogeneity is significant (*p* = 0.001), and more than 76% of the variability comes from heterogeneity (Figure 8). The funnel plot does not suggest publication bias (Figure 9).

#### 3.3.3. Meta-Analysis of In-Vitro Studies of Peak Load for Pull-Out Strength Regarding Thread Shape Factor

(A)artificial bone model

There is small significant (*p* = 0.027) negative effect size. Study results are inconsistent—heterogeneity is significant (*p* < 0.001), and more than 97% of the variability comes from heterogeneity (Figure 10). The funnel plot confirms high heterogeneity, asymmetry suggests some publication bias (Figure 11).

(B)animal bone model

There is small significant (*p* = 0.033) positive effect size. Study results are inconsistent—heterogeneity is significant (*p* = 0.030), and more than 67% of the variability comes from heterogeneity (Figure 12). The funnel plot confirms high heterogeneity, asymmetry suggests some publication bias (Figure 13).

## 4. Discussion

Although there are many systematic reviews concerning different geometric characteristics of OMIs, the present paper is the first referring to their minidesign. First of all, it should be pointed out that no clinical study considering the number of threads, threads depth, or TSF has been found in the literature. The only papers found correlating the minidesign of OMIs to their physical characteristics are animal studies, in-vitro studies, and 3D finite element analysis. Thus, the authors of the present systematic review must have based their clinical recommendations on indirect evidence.

Moreover, there is a well-designed split-mouth study discussing other factors, that may influence OMI stability, for instance, chemical treatment of screw surfaces [44]. Another study shows that increasing penetration depth of OMIs results in better retention [45], whereas increased abutment head distance from cortical plate leads to decreased retention [45]. Moreover, TADs inclination angle of 60 to 70° to the cortical plate was reported as the most retentive insertion angle [46,47]. Insertion at a right angle or more oblique from the line of force reduces retention of TADs.

In the present review with meta-analysis, the studies included are characterized mainly by the medium quality of evidence. This may result from the types of studies included. In the studies included the researchers had a possibility to carefully design every step of a trial, including the study material, procedure, and examination, leaving less possibility of biases than in clinical trials, where the subjects may much more frequently behave differently than planned. The shortcomings of in-vitro studies included were mainly lack of sample size calculation, randomization, and blinding of the results evaluation. Some of the studies have only discussed the proportions of the ongoing phenomena, without providing specific values. The present meta-analysis provides very interesting results of the calculated effect. The high heterogeneity could not have been disclosed. Various implants were tested in different environments, including a digital analysis environment. A similar problem was observed in another recently published meta-analysis on OMIs [48]. Therefore, the studies included were classified according to their design in terms of the environment. Moreover, studies included had to meet the requirement of including minimum of three study groups of subjects for comparison (required for meta-analysis). It is surprising that all the studies carried out on the artificial bone model are characterized by an enormous heterogeneity, and all funnel plots are indicating publication bias. On the contrary, studies performed on the animal model are characterized by lower I^2^ and funnel plots do not indicate bias. Additionally, the results of artificial bone model studies are in contradiction to studies performed on an animal model. Both studies on the artificial bone model and those on the animal model indicate that smaller thread pitch is correlated with higher pull-out strength values. However, the results in the animal model are more homogeneous. Referring to studies included only in the systematic review, Topcuoglu et al. found that a smaller thread pitch is correlated with higher removal torque values [25]. A smaller thread pitch prevents lateral displacement when orthodontic force is exerted [30]. The same was indicated by Budsabong et al. in their in-vitro study [43]. This fact additionally strengthens the result of the meta-analysis. Dastenaei et al. in a 3D finite element analysis point out that thread pitch also increases the stability of OMIs, but with thread pitch density that is too high, the implant may be more prone to fracture [28]. Similar results were observed regarding thread depth. In both groups of studies, it was stated that thread depth does not significantly influence pull-out strength, e.g., the primary stability of the implant. However, also here the results on the animal model are more homogeneous. There is no clear indication regarding TSF. Studies on the artificial bone model stated that smaller TSF was correlated with better stability, whereas studies on the animal bone model stated that bigger TSF was correlated with better stability. Both groups are characterized by high I^2^ and the funnel plot suggests publication bias, thus there is no clear scientific evidence regarding an optimal TSF. In the studies regarding MIT, it was found that a smaller thread pitch was correlated with higher MIT values. What is worth mentioning is that in all studies included examined OMIs were made of titanium grade V which is proof of the wide recognition of this material in orthodontics [49]. An important factor that may affect the clinical effectiveness of OMI is also its head, which should be accessible to the clinician and make fixing wire or elastics easy and efficient (e.g., button for the elastics, slot for wire) [50]. An interesting element that should also be taken into account is the thread shape. Gracco et al. [32] indicate reverse butter shape, and Yashwant [40]—trapezoidal fluted and reverse butter shape as shapes ensuring the greatest stability. Clear conclusions can be drawn from the review that clinicians should use OMIs with smaller thread pitch to obtain maximum anchorage with high primary stability and avoid OMIs lateral displacement during therapy. However, one should remember that excessive thread pitch (<0.45mm) may cause significant strain within the bone due to sparse stress distribution and disturb the physiological bone remodeling process. Minidesign features such as thread depth or TSF do not seem to be very clinically significant in view of the available knowledge. Another interesting factor could be the collar shape. Clinically, implants with a wider neck exhibit better long-term retention [51]. Miniscrew design could be of special importance in patients, who might have compromised miniscrew retention resulting from previous treatment influencing bone metabolism, including chemotherapy [52]. Finally, it should be mentioned that digital treatment planning, mesh superimposition (intraoral scans + CBCT), and CAD-Cam technologies, including CBCT, and planned guided insertion, may be other important factors influencing miniscrew stability [53,54]. It has been proven that implants inserted through 3D guides were characterized by better stability [55].

The limitations of the present study come from the number of studies present in the literature, the difference in study design, different results reporting, and a lack of proper clinical studies. More studies are needed in the future to accurately detect and determine the effect size of a given minidesign characteristic.

## 5. Conclusions

Minidesign of orthodontic mini-implant—that is characteristics such as OMI thread pitch, OMI thread depth, and OMI thread shape should be considered when choosing optimal miniscrews for orthodontic anchorage.Thread pitch seems to be of special importance for OMIs retention—OMIs with a more dense thread—should be preferred due to their superior stability.Thread depth seems to be of low importance for OMIs stability.There is no clear scientific evidence referring to the optimal tread shape factor.Studies present in the literature vary greatly in study design and way of reporting results. The results of in-vitro tests carried out on animal models are more consistent than those carried out on artificial bone models.

## Figures and Tables

**Figure 1 jcm-11-05304-f001:**
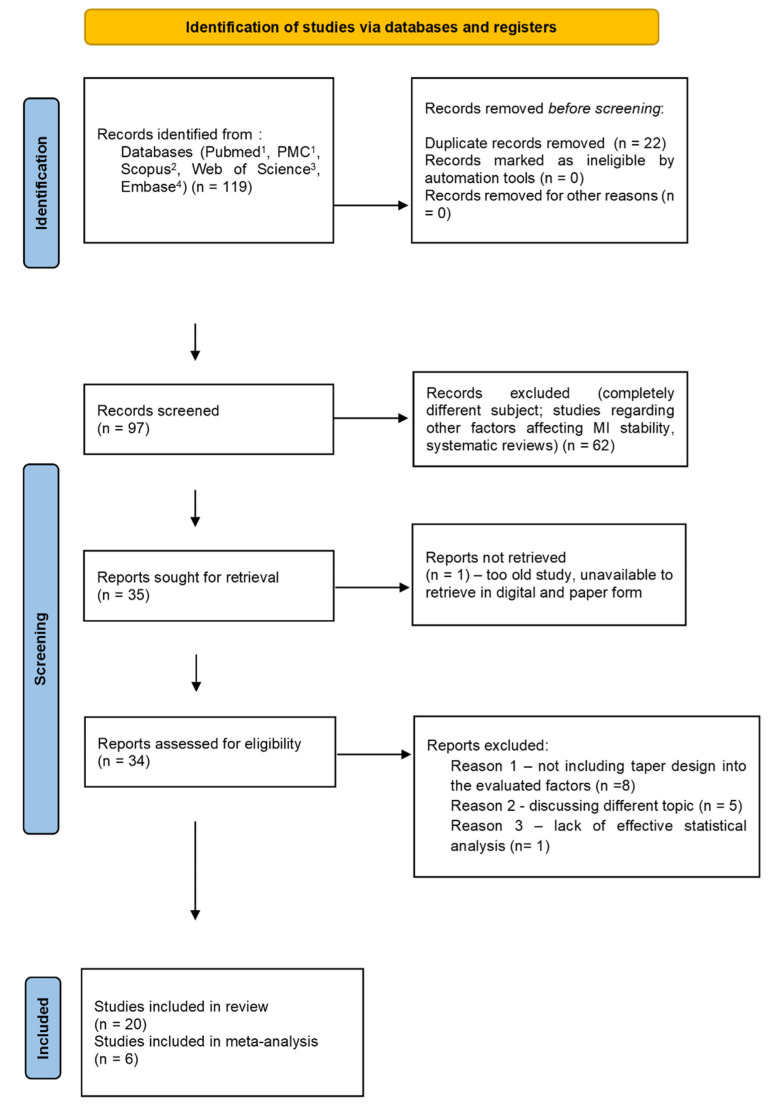
Prisma 2020 flow diagram. ^1^—search string: (“mini-implant” OR “miniscrew” OR “TAD” OR “temporary anchorage device” OR “skeletal anchorage”) AND (“orthodontics” [MeSH Terms] OR “malocclusion” [MeSH Terms] OR “Tooth Movement Techniques” [MeSH Major Topic]) AND (“thread” OR “thread pitch” OR “thread depth” OR ”thread shape” OR “thread design”)AND (“success rate” OR “success” OR “successful” OR “survival rate” OR “failure risk” OR “treatment success” OR “stability”)—20 + 45 results; ^2^—search string: TITLE-ABS-KEY ((“mini-implant” OR “miniscrew” OR “TAD” OR “temporary anchorage device” OR “skeletal anchorage”) AND (“orthodontics” OR “malocclusion” OR “Tooth Movement Techniques”) AND (“thread” OR “thread pitch” OR “thread depth” OR “thread shape” OR “thread design”) AND (“success rate” OR “success” OR “successful” OR “survival rate” OR “failure risk” OR “treatment success” OR “stability”))—24 results; ^3^—search string: Results for (“mini-implant” OR “miniscrew” OR “TAD” OR “temporary anchorage device” OR “skeletal anchorage”) AND (“orthodontics” OR “malocclusion” OR “Tooth Movement Techniques”) AND (“thread” OR “thread pitch” OR “thread depth” OR “thread shape” OR “thread design”) AND (“success rate” OR “success” OR “successful” OR “survival rate” OR “failure risk” OR “treatment success” OR “stability”) [All fields]—18 results; ^4^ —search string: (‘mini-implant’ OR ‘miniscrew’/exp OR ‘miniscrew’ OR ‘tad’ OR ‘temporary anchorage device’/exp OR ‘temporary anchorage device’ OR ‘skeletal anchorage’) AND (‘orthodontics’/exp OR ‘orthodontics’ OR ‘malocclusion’/exp OR ‘malocclusion’ OR ‘tooth movement techniques’/exp OR ‘tooth movement techniques’) AND (‘thread’/exp OR ‘thread’ OR ‘thread pitch’ OR ‘thread depth’ OR ‘thread shape’ OR ‘thread design’) AND (‘success rate’/exp OR ‘success rate’ OR ‘success’/exp OR ‘success’ OR ‘successful’ OR ‘survival rate’/exp OR ‘survival rate’ OR ‘failure risk’ OR ‘treatment success’/exp OR ‘treatment success’ OR ‘stability’/exp OR ‘stability’) AND [embase]/lim—12 results.

**Figure 2 jcm-11-05304-f002:**
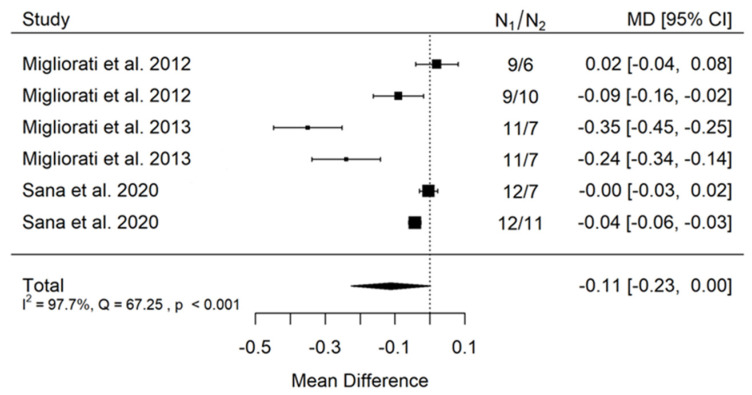
Forest plot for evalutaion of thread pitch dimension infulence on pull-out strentght in artifical bone model studies [33,34,41].

**Figure 3 jcm-11-05304-f003:**
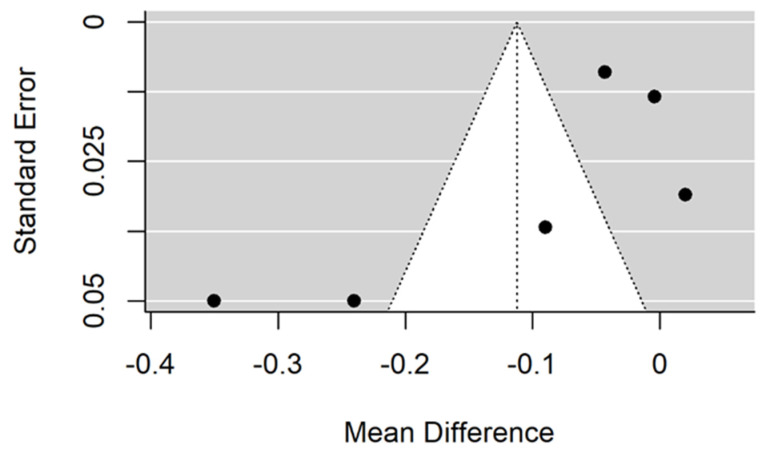
Funnel plot for evalutaion of thread pitch dimension infulence on pull-out strentght in artifical bone model studies.

**Figure 4 jcm-11-05304-f004:**
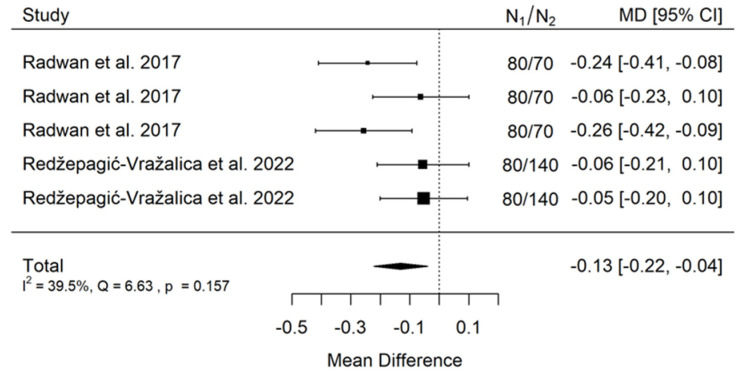
Forest plot for evalutaion of thread pitch dimension infulence on pull-out strentght in animal bone model studies [38,44].

**Figure 5 jcm-11-05304-f005:**
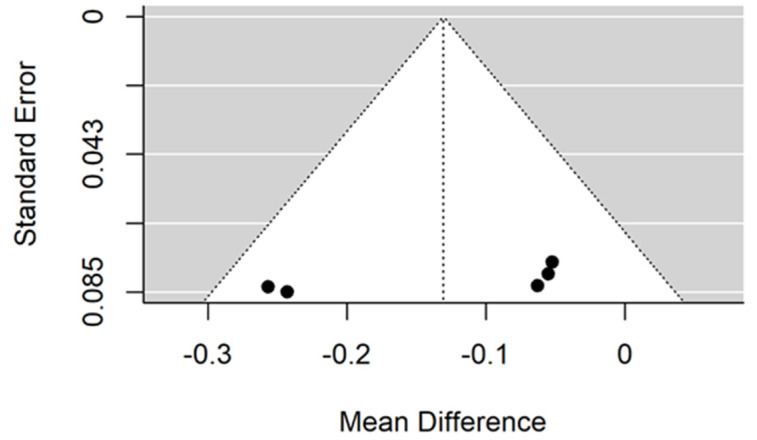
Funnel plot for evalutaion of thread pitch dimension infulence on pull-out strentght in animal bone model studies.

**Figure 6 jcm-11-05304-f006:**
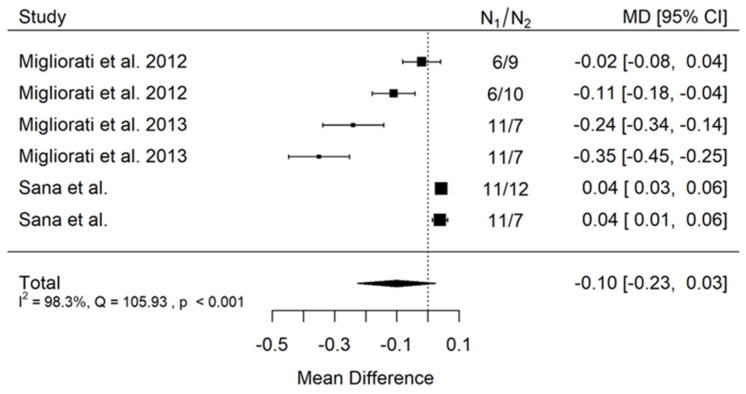
Forest plot for evalutaion of thread depth dimension infulence on pull-out strentght in artifical bone model studies [33,34,41].

**Figure 7 jcm-11-05304-f007:**
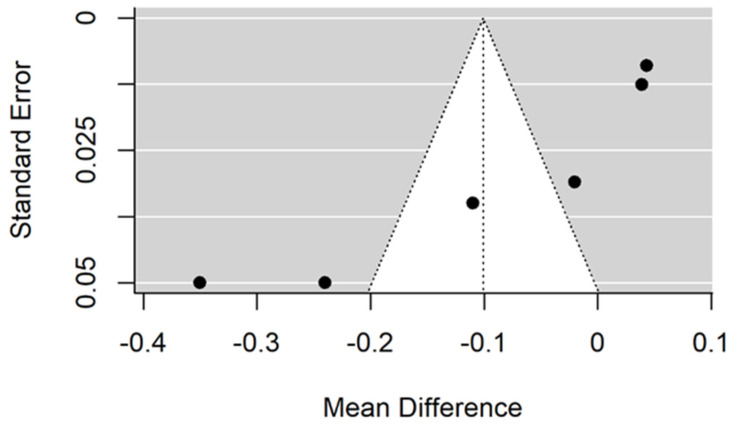
Funnel plot for evalutaion of thread depth dimension infulence on pull-out strentght in artifical bone model studies.

**Figure 8 jcm-11-05304-f008:**
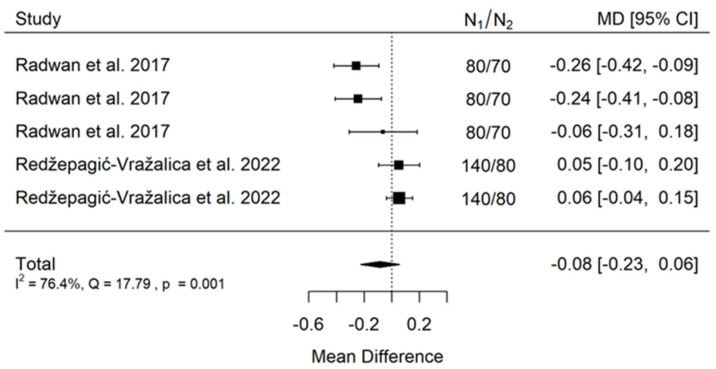
Forest plot for evalutaion of thread depth dimension infulence on pull-out strentght in animal bone model studies [38,44].

**Figure 9 jcm-11-05304-f009:**
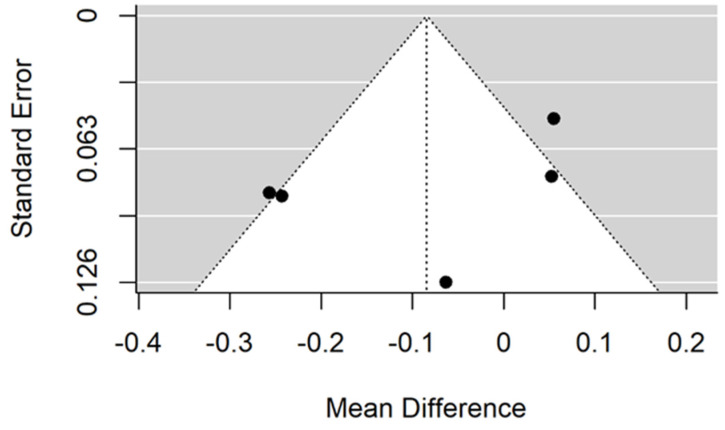
Funnel plot for evalutaion of influence of thread depth dimension on pull-out strentght in animal bone model studies.

**Figure 10 jcm-11-05304-f010:**
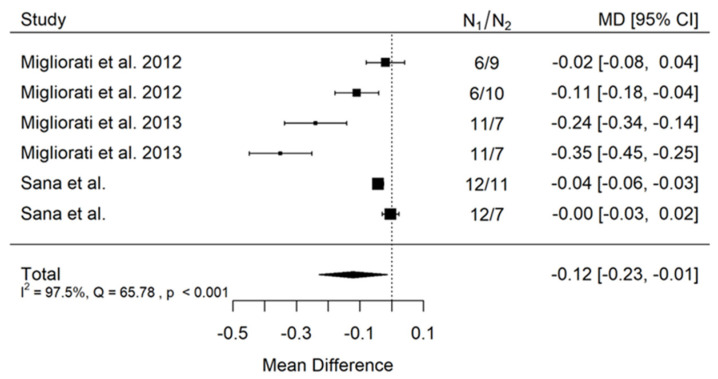
Forest plot for evalutaion of thread shape factor infulence on pull-out strentght in artifical bone model studies [33,34,41].

**Figure 11 jcm-11-05304-f011:**
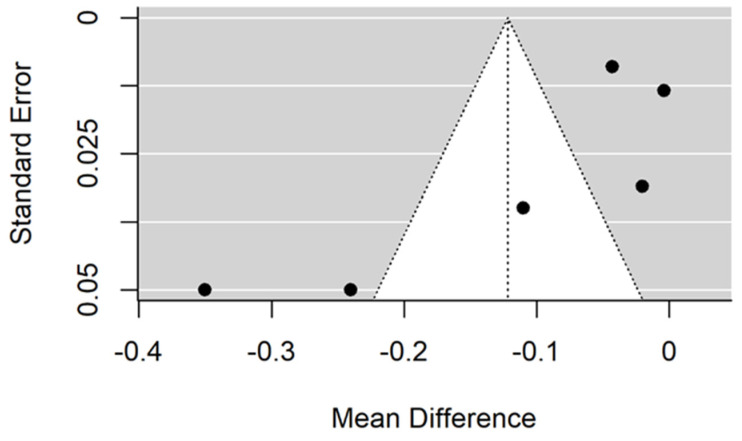
Forest plot for evalutaion of thread shape factor influence on pull-out strentght in artifical bone model studies.

**Figure 12 jcm-11-05304-f012:**
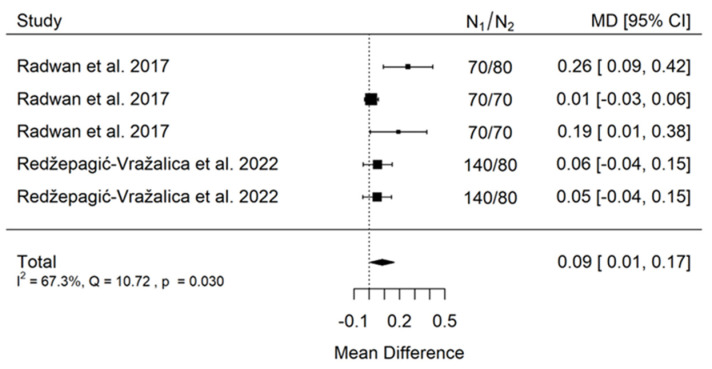
Forest plot for evalutaion of thread shape factor infulence on pull-out strentght in animal bone model studies [38,44].

**Figure 13 jcm-11-05304-f013:**
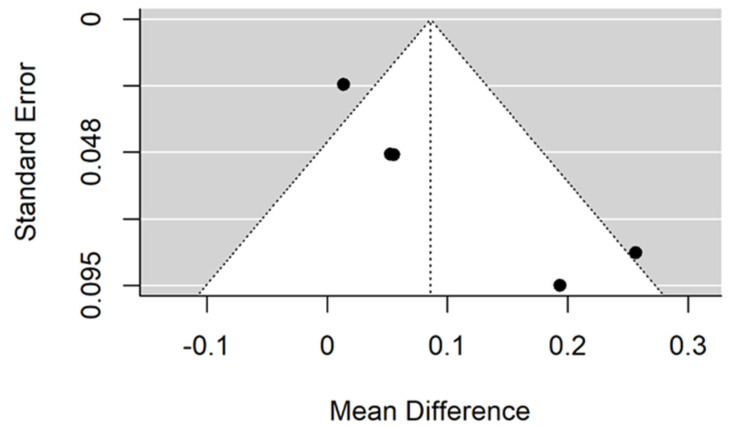
Funel plot for evalutaion of thread shape factor infulence on pull-out strentght in animal bone model studies.

**Table 1 jcm-11-05304-t001:** Characteristics of included studies.

Author and Year	Type of Article	Study Groups	Method	Outcome Measured	Results
Kim et al. 2011 [25]	Prospective randomized animal study with in-vitro analysis	24 tapered mini-implants of 5 mm length 24 cylindrical mini-implants of 7 mm length	Insertion and removal torque test in rabbit bone	Insertion and Removal torque values [N/cm]	Despite the fact that the tapered implants were shorter, they presented higher torque values than cylindrical at each time of the test.
Topcuoglu et al. 2013 [26]	Prospective randomized animal study with in-vitro analysis	4 groups of 20 immediately loaded Ti6Al4V OMIs with interpitch distance of I—0.694 mmm, II—0.721 mm, III—0.693 mm and IV—0.702 mm inserted into rabbit fibula 4 groups of 20 unloaded Ti6Al4V OMIs with interpitch distance of I—0.694 mmm, II—0.721 mm, III—0.693 mm and IV—0.702 mm inserted into rabbit fibula	Removal torque test, SEM, and histomorphometric analyses in rabbit bone	Removal torque values [N/cm]	More frequent thread pitch had a positive effect on stability in every group studied. I—8.50 (2.41–10.05) 8.10 (4.94–9.35) II—6.92 (2.76–8.48) 4.63 (3.53–8.59) III—6.27 (3.99–9.87) 4.59 (2.26–5.57) IV—5.78 (4.17–7.95) 4.10 (2.59–5.53)
Chang et al. 2012 [27]	3D finite element analysis	Four types of titanium grade V OMIs with different design parameters Screw type 1 had a 0.4-mm thread depth and a 7 tapered core at the 5 uppermost threads. Screw type 2 had a 0.4-mm thread depth and a 0 tapered core (cylindrical core). Screw type 3 had a 0.4-mm thread depth and a 7 tapered core at the 3 uppermost threads. Screw type 4 had a 0.32-mm thread depth and a 7 tapered core at the 3 uppermost threads. To evaluate the effect of thread depth on primary stability, the thread depths of the mini-implants were set at 0.16, 0.24, 0.32, 0.40, and 0.48 mm;	Assessment of possible mini-implant insertion torque stress and of displacement within the trabecular bone	Maximum insertion torque [Ncm], Pullout strength (N), Displacement before failure (mm), Stiffness (N/mm)	Mini-implants with greater thread depths, smaller tapers, and shorter taper lengths generated higher maximum stresses on the bone and thread elements. These mini-implants had larger relative displacements, as well. Pullout resistance increased as thread depth increased from 0.16 to 0.32 mm. However, the pullout resistance decreased as thread depth exceeded 0.32 mm. Pullout resistance also decreased as taper degrees and taper lengths decreased. High stresses were distributed on the uppermost threads at the neck of the mini-implants close to the bone margin in all conditions. Maximum insertion torque was observed in the first and the third OMIs.
Shen et al. 2014 [28]	3D finite element analysis	9 samples of titanium grade V mini-implants of thread heigh of 0.1 mm, 0.25 mm and 0.4 mm height and thread pitch of 0.5 mm, 1.20 mm, and 2 mm.	3D finite element analysis of type III bone according to the Lekholm & Zarb classification (maxillary posterior region)	Maximum equivalent stresses	Increased thread height with a thread pitch of 1.20 mm was superior for the maxillary posterior region. Thread height proved more important for reducing maxillary stress and enhancing orthodontic mini-implant stability than the thread pitch.
Dastenaei et al. 2015 [29]	3D finite element analysis	Titanium grade V OMI of diameter 1.6 mm, length 8 mm, thread pitch of 0.75 mm, thread depth of 0.25 mm and thread height of 0.331, tip angle 63°	Assessment of stress points within the implant inserted to trabecular bone	Maximum equivalent stresses	Stress concentration usually occured at the first thread of the implant. Stress decreased when screw pitch decreased from 1 to 0.5 mm; it was concentrated at the apex of the threads. The stress increased when the screw pitch became less than 0.45 mm and the stress distribution pattern was sparse. It seems appropriate to create a new dual miniscrew design that can provide ergonomic aims, with bigger thread pitch at the apex and smaller on threads neck.
Pouyafar et al. 2021 [30]	3D finite element analysis	Series of test leading to optimal design of OMI	Assessment of possible mini-implant displacement within the trabecular bone	The lateral displacement measurement [µm]	The conical section improved the initial stability by creating compressive stress and additional friction in the surrounding bone. With increasing each millimeter and each degree in the conical section’s length and angle, the lateral displacement decreased by 2.3 and 1.8mm, respectively. The length and angle of the non-threaded part does not significantly control the lateral displacement. The higher the pitch of the mini-implant, the higher the lateral displacement (increases by 1.3– 2.2mm). It is necessary to consider the minimum possible value for the pitch according to the threads’ shape.
Kim et al. 2009 [31]	In-vitro study	Titanium grade V (10 of each group) 6 mm cylindrical, 8 mm cylindrical, 6 mm taper, 8 mm taper, 6 mm dual-thread and taper and 8 mm dual-thread and taper	Mechanical assessment of toque in artificial bone block from polyurethane foam block	Insertion torque (MIT), maximum removal torque (MRT), torque ratio (TR; MRT/MIT), insertion angular momentum (IAM), removal angular momentum (RAM)	The removal torque of the taper shape was lower than the removal of torque of the dual-thread shape. The dual-thread shape showed a low insertion torque and a gentle increase of insertion torque. The dual-thread shape also showed a higher removal torque on the broad range than the cylindrical and taper shapes. Long mini-implants need higher insertion torque than short mini-implants. Dual-thread shape may need improvement for reducing the long insertion time to decrease the stress to the surrounding tissue.
Gracco et al. 2012 [32]	In-vitro study	35 OMIs (7 in each group) five different designs in thread shape (reverse buttress, buttress, 75° joint profile with flutes, trapezoidal and rounded)	Pull out strength form artificial bone block from polyurethane foam block	Pull out strength [N]	The thread shape influenced the resistance to pullout and, therefore, the primary stability of miniscrews. The buttress reverse thread shape (about 192.8 N) had consistently higher pullout strength values than the other designs. They fared worse in turn rounded, trapezoidal, 75° joint profile design and buttress profile)
Migliorati et al. 2012 [33]	In-vitro study	Three types of OMIs—one stainless steel, two titanium grade V of thread depth of 0.1735 mm, 0.1926 mm, 0.2757 mm and thread pitch of 0.9172 mm, 0.8255 mm and 0.1043 mm	Mechanical assessment of toque and screw mobility in artificial bone block with different densities from polyurethane foam	Peak load at the pull out tests [KN]	Increased thread shape factor (depth/pitch) is positively correlated with resistance to extraction.
Migliorati et al. 2013 [34]	In-vitro study	30 titanium grade V OMIs in 3 groups: (10 in each group) of thread depth of 0.345, 0.216, 0.114mm and thread pitch of 0.826, 0.894, 0.574	Mechanical assessment of toque and screw mobility in artificial bone block with different densities from polyurethane foam	Maximum insertion torque [Ncm], peak load at the pull out tests [KN]	There is a direct positive correlation between the increase in TSF (depth/pitch), the miniscrew pull-out strength and maximum insertion torque
Da Cunha et al. 2015 [35]	In-vitro study	20 titanium grade V OMIs in 2 groups (10 in each group) G1 of thread pitch 30 × 0.6 mm and G2 45 × 0.8 mm	Mechanical assessment of torque and screw mobility in artificial bone block with different densities of polyurethane foam	Maximum insertion torque, removal torque, loss of torque [Ncm], implant stability [x]	The mini-implants with a shorter pitch distance and an insertion angle of 30° presented a better primary stability (torque) in artificial bone of a higher density. The mini-implants with a longer pitch distance and an insertion angle of 45° were found to be more stable in artificial bone of lower density, when performing evaluation with the Periotest.
Walter et al. 2013 [36]	In-vitro study	240 self-drilling titanium Titanium Grade V OMIs of 12 types from 8 manufacturers **The authors did not provide exact geometrical characteristics for every type od the screw—only correlations**	Mechanical tests of pull-out strength, torsional fracture and insertion torque in artificial bone block of polyurethane foam, SEM inspection	Maximum insertion torque, Torsional Fracture [Ncm], pull-out strength [N]	Within the medium diameter OMI group, conical screws had higher insertion torque and torsional fracture values than cylindrical OMIs. Greater thread depth was related to higher pull-out strength values, although OMIs with similar pull-out strength values may have different insertion torque values. Thread depth and pitch had some impact on pull-out strenght. Torsional fracture depended mainly on the OMI inner and outer diameters. A thread depth to outer diameter ratio close to 40% increased torsion fracture risk.
Cha et al. 2015 [37]	In-vitro study	Titanium grade V OMIs with single-thread (thread pitch 0.7) and dual-thread design (lower part pitch 0.7 and 0.35 upper pitch part)	Insertion torque test and strain of bone-implant interface in artificial bone block from polyurethane foam block	Strain in [µstrains] and Insertion torque in [Nm]	The strain between the single-thread and dual-thread type miniscrews was similar at a cortical bone thickness of 1.0mm, but the discrepancy between miniscrew types widened to >10,000 μstrain with increasing cortical bone thicknesses. Self-drilling dual-thread miniscrews provide better initial mechanical stability, but their design may cause excessive strain that is over the physiological bone remodeling level (>1mm cortical bone) at the bone-implant interface of thick cortical bone layers.
Radwan et al. 2017 [38]	In-vitro study	40 orthodontic miniscrews of same diameter and length and different taper designs randomly inserted into pilot holes	Mechanical assessment of pull-out test and periotest in 10 embalmed human maxillae	Pull-out strength estimation [N], implant stability [x]	Larger pitch width, flank, thread angle, apical face angle, and/or lead angle led to a higher primary stability, while a smaller thread shape factor (depth/width) improved primary stability.
Katić et al. 2017 [39]	In-vitro study	In each tested group there were 10 cylindrical self-drilling titanium grade 5 Ortho Easy^®^ (FORESTADENT^®^, Pforzheim, Germany), 1.7 × 6 mm and 1.7 × 8 mm; Aarhus Anchorage System (MEDICON eG, Tuttlingen, Germany), 1.5 × 6 mm and 1.5 × 8 mm; and Jeil Dual Top™ Anchor System (Jeil Medical Corp., Seoul, Korea), 1.4 × 6 mm, 1.6 × 6 mm, 2.0 × 6 mm, 1.4 × 8 mm, 1.6 × 8 mm, and 2.0 × 8 mm	Maximum insertion torque test in rabbit bone	Maximum insertion torque [Nmm], Vertical Force [N], Torsion [Nmm]	The multiple linear regression model showed that significant predictors for higher maximum insertion torque were: a larger implant diameter, a higher insertion angle, and thicker cortical bone. Manufacturers should consider increasing the insertion angle of the implant to improve the implant design and achieve a better primary stability in cases where the operator cannot use a larger implant diameter.
Yashwant et al. 2017 [40]	In-vitro study	50 OMIs (10 in each group) of five different designs in thread shape (reverse buttress, buttress, 75° joint profile with flutes, trapezoidal and trapezoidal fluted)	Pull out strength in artificial bone block from polyurethane foam block	Pull out strength [N]	Trapezoidal fluted mini implants showed twice as higher pull out strength then mini implants of other thread designs used in this study (about 61N), followed by reverse buttress (about 27N), (buttress, 75° joint profile with flutes) which showed similar values (about 26N). A trapezoidal design presented the lowest value (13N).
Sana et al. 2020 [41]	In-vitro study	3 Titanium grade V OMIs— ORTHOImplant (3M Unitek, Monrovia, CA, USA): 1.8-mm diameter and 8-mm length, TOMAS (Dentaurum): 1.6-mm diameter and 8-mm length, VECTOR TAS (Ormco): 1.4-mm diameter and 8-mm length.	Pull out strength in artificial bone block from polyurethane foam block	Pull out strength [N]	Orthoimplant type with a larger diameter, smaller pitch and shorter taper length has a better primary stability, and lower stresses within the mini-implants and surrounding comparing to other groups tested. The favorable insertion angulation found was 90°, as it provides better primary stability and low stresses in the mini-implant and surrounding bone under orthodontic loading.
Watanbe et al. 2022 [42]	In-vitro study	Cylindrical vs. classic thread shape vs. novel thread shape Titanium grade V OMIs **The authors did not provide exact geometrical characteristics for every type od the screw—only correlations**	Mechanical assessment of toque, screw mobility and stiffness in artificial bone block from polyurethane foam	Values of maximum insertion torque, removal torque, torque ratio, [Ncm} screw mobility, [mm] static stiffness, dynamic stiffness [N/mm} and energy dissipation	Compared to miniscrews of a regular thread shape, the novel miniscrew of a different thread shape showed a higher torque ratio, a lower stiffness and screw mobility. These features seem important for the initial stability.
Budasbong et al. 2022 [43]	In-vitro study	60 custom made titanium Grade V OMIs	Mechanical assessment of toque in 10 embalmed human maxillae	Maximum insertion torque [Nm], implant stability [x]	The maximum insertion torque and implant stability tests demonstrated a pitch-dependent decrease. The pitch had a strong negative correlation with maximum insertion torque and implant stability, while the cortical bone thickness had a strong positive correlation with these outcomes.
Redžepagić-Vražalica et al. 2022 [44]	In-vitro study	40 Titanium grade V OMIs of which 20	Mechanical assessment of toque in 40 pork ribs	Maximum insertion torque [Nm], Pull out strength [N]	The design of the mini-implant affects the insertion torque and pulling force. The bone quality at the implant insertion point is important for primary stability; thus, the increase in the cortical bone thickness significantly increases the pulling force.

**Table 2 jcm-11-05304-t002:** Quality assessment according to SYRCLE Risk Assessment Tool.

Item	Type of Bias	Domain	Kim et al. 2011 [25]	Topcuoglu et al. 2013 [26]
1	Selection bias	Sequence generation	-	+
2	Selection bias	Baseline characteristics	+	+
3	Selection bias	Allocation concealment	?	+
4	Performance bias	Random housing	?	?
5	Performance bias	Blinding	-	?
6	Detection bias	Random outcome assessment	-	+
7	Detection bias	Blinding	-	?
8	Attrition bias	Incomplete outcome data	+	+
9	Reporting bias	Selective outcome reporting	+	+
10	Other	Other sources of bias	?	?

**Table 3 jcm-11-05304-t003:** Quality Assessment according to Methodological Quality Assessment of Single-Subject Finite Element Analysis Used in Computational Orthopaedics (MQSSFE).

	Question	Chang et al. 2012 [27]	Shen et al. 2014 [28]	Dastenaei et al. 2015 [29]	Pouyafar et al. 2021 [30]
Study Design and Presentation of Findings					
1	Was the hypothesis/aim/objective of the study clearly described?	Yes	Yes	Yes	Yes
2	Were all analyses planned at the outset of study? *Answer NO for unplanned analysis/sub-analysis, unable to determine.*	Yes	Yes	Yes	Yes
3	If data dredging (establish objectives, hypothesis and endpoint parameters without scientific reason) was used, was the spectrum of the data justified by any concepts? *Answer YES if no data dredging, NO if unable to determine*	Yes	Yes	Yes	Yes
4	Were ALL the outcome measures and parameters (including all data reduction methods or derived parameters) clearly described and defined in the Objectives or Methods section? *Answer NO if they are only defined in results or discussion*	Yes	Yes	No	No
5	Were the time points or period for ALL the outcome measures clearly described? *Answer YES if not applicable*	Yes	No	No	Yes
6	Were the main outcome measures appropriate to describe the targeted conditions? *Answer NO if unable to determine*	Yes	Yes	Yes	Yes
7	Were the key findings described clearly?	Yes	Yes	Yes	Yes
8	Were ALL the contour plots that were used for comparison presented with the same colour scale?	Yes	Yes	Yes	Yes
Subject Recruitment					
9	Were the characteristics of the model subject clearly described?	Yes—implants with features invented by researchers	Yes—specific implant available for purchase	Yes—specific implant available for purchase	Yes—specific implant available for purchase
10	Were the principal confounders of the model subject clearly described? (Age, sex, or body weight, and height)	Yes—standard artificial bone features	Yes—standard artificial bone features	Yes—standard artificial bone features	Yes—standard artificial bone features
11	Was the model subject participated in the study representative of the population with the targeted clinical conditions or demographic features? (e.g., answer NO if simulating a pathology by modifying a normal subject model; or scaling an adult model to a child model)	Yes—standard artificial bone features	Yes—standard artificial bone features	Yes—standard artificial bone features	Yes—standard artificial bone features
12	Were the targeted intervention or clinical condition clearly described? (with details in the severity, class, design/dimensions of implants, or details in surgical surgery)	Yes—standard artificial bone features	Yes—standard artificial bone features	Yes—standard artificial bone features	Yes—standard artificial bone features
Model Reconstruction and Configuration					
13	Was the model reconstruction modality for the body parts and ALL other items, such as implants, clearly described (e.g., MRI, 3D-scanning, CAD)?	Yes	Yes	Yes	Yes
14	Were ALL important technical specifications (e.g., resolution) for the reconstruction modality clearly described?	Yes	Yes	Yes	Yes
15	Was the posture or position of the body parts controlled during the acquisition process (e.g., MRI, CT) of the model reconstruction?	Yes/No	Yes	Yes	Yes
16	Were the model reconstruction methods for ALL components clearly described including those requiring additional procedures (e.g., connecting points for drawing ligaments from MRI)?	Yes/No	Yes	Yes	Yes
17	Were the orientation or relative position among the components of the model assembly (where appropriate) clearly described? *Answer YES if not applicable*	Yes	Yes	Yes	Yes
18	Was the type of mesh for ALL components, including the order of magnitude of the elements, clearly described?	No	No	No	Yes
19	Were the material properties for ALL components clearly described and justified? (e.g., with reference)	Yes	Yes	No	Yes
20	Were ALL the contact or interaction behaviours in the model clearly described and justified?	Yes/No	Yes/No	No	No
Boundary and Loading Condition (Simulation)					
21	Were the boundary and loading conditions clearly described?	Yes	Yes	Yes	Yes
22	Was the boundary and loading condition sufficiently simulating the common activity/scenario of the conditions? (e.g., if the research or inference is targeted to ambulation or daily activities, simulations of balanced standing or pre-set compressive load are insufficient)	Yes	Yes	Yes	Yes
23	Was the model driven by the boundary condition acquired from the same model subject?	Yes	Yes	Yes	Yes
24	Was loading condition on the scenario sufficiently and appropriately considered in the simulation? (e.g., muscle force, boundary force, inertia force)	Yes	Yes	Yes	Yes
25	Was the loading condition acquired from the same model subject?	Yes	Yes	Yes	Yes
26	Were the software (e.g., Abaqus, Ansys), type of analysis (e.g., quasi-static, dynamic), AND solver (e.g., standard, explicit) clearly described? (solver can be regarded as clearly described if it is obvious to the type of analysis)	Yes	Yes	Yes	Yes
Model Verification and Validation					
27	Were the methods of mesh convergence or other verification tests conducted and clearly described?	No	Yes	No	Yes
28	Were the model verification conducted and results presented clearly; and that the model was justified acceptable?	No	Yes	No	No—just mentioned about carrying them out
29	Were direct model validation (with experiment) conducted and described clearly? *Answer YES if the authors had direct model validation previously with reference.*	No	Yes	No	Yes
30	Were the model validation conducted and results presented clearly; and that the model was justified acceptable?	No	Yes	No	Yes
31	Were the model prediction or validation findings compared to relevant studies?	No	Yes	Yes	No
Model Assumption and Validity					
32	Were the model assumptions or simplifications on model reconstruction/configuration AND material properties discussed?	No	Yes	No	Yes
33	Were the model assumptions or simplifications on the boundary and loading conditions discussed?	Yes/No	Yes	Yes/No	Yes
34	Were the limitations of model validation discussed? (e.g., differences in case scenario; differences between validation metric and primary outcome)	Yes	No	No	No
35	Was the limitation on external validity, single-subject, and subject-specific design discussed?	No	Yes	No	Yes
36	Were there any attempts to improve or discuss internal validity (such as mesh convergence test), uncertainty and variability in the study?	No	Yes	Yes	Yes
37	Was there any discussion, highlights or content on the implications or translation potential of the research findings? *Answer NO if there are only bold claims without making use of the result findings or key concepts*	No	Yes	No	Yes
Sum:		26	34.5	23.5	33

**Table 4 jcm-11-05304-t004:** Quality assesment of in-vitro studies according to QUIN assesment tool.

Criteria No.	Criteria	Kim et al. 2009 [31]	Gracco et al. 2012 [32]	Migliorati et al. 2012 [33]	Migliorati et al. 2013 [34]	Da Cunha et al. 2015 [35]	Walter et al. 2013 [36]	Cha et al. 2015 [37]
1	Clearly stated aims/objectives	2	2	2	2	2	2	2
2	Detailed explanation of sample size calculation	0	0	0	0	2	0	0
3	Detailed explanation of sampling technique	2	2	2	2	1	2	2
4	Details of comparison group	2	2	1	2	2	2	2
5	Detailed explanation of methodology	2	1	1	1	2	2	2
6	Operator details	0	0	1	2	2	2	2
7	Randomization	0	0	0	0	0	0	0
8	Method of measurement of outcome	2	1	2	2	2	1	2
9	Outcome assessor details	1	2	2	2	1	1	1
10	Blinding	0	0	0	0	0	0	0
11	Statistical analysis	2	2	2	2	2	2	2
12	Presentation of results	2	2	1	2	1	1	2
Criteria No.	Criteria	Radwan et al. 2017 [38]	Katić et al. 2017 [39]	Yashwant et al. 2017 [40]	Sana et al. 2020 [41]	Watanbe et al. 2021 [42]	Budasbong et al. 2022 [43]	Redžepagić-Vražalica et al. 2022 [44]
1	Clearly stated aims/objectives	2	2	2	2	2	2	2
2	Detailed explanation of sample size calculation	0	0	0	0	0	0	0
3	Detailed explanation of sampling technique	1	2	1	2	2	2	2
4	Details of comparison group	1	2	1	2	2	2	2
5	Detailed explanation of methodology	1	2	1	1	2	2	1
6	Operator details	1	2	1	2	2	2	0
7	Randomization	0	0	0	0	0	0	0
8	Method of measurement of outcome	2	2	1	2	1	1	2
9	Outcome assessor details	2	2	1	2	2	1	0
10	Blinding	0	0	0	0	0	0	0
11	Statistical analysis	2	2	2	2	2	2	2
12	Presentation of results	1	2	1	2 *—additonally performed FEA does not bring any sigificant information to the study	1	2	2

**Table 5 jcm-11-05304-t005:** Meta-analysis of in-vitro studies of peak load for pull-out strength—artificial bone model.

Author	Number of Implants with Reduced Thread Pitch (Number of Threads-Observations)	Dimension of Thread Pitch [mm]	Pull-Out Strength [KN]	Number of Implants with Greater Thread pitch (Number of Threads-Observations)	Dimension of Thread Pitch [mm]	Pull-Out Strength [KN]
Migliorati et al. 2012 [33]	1 (9)	0.8255 ± 0.0282	0.34 ± 0.07	1 (6)	0.9172 ± 0.0655	0.32 ± 0.05
Migliorati et al. 2012 [33]	1 (9)	0.8255 ± 0.0282	0.34 ± 0.07	1 (10)	1.043 ± 0.0306	0.43 ± 0.09
Migliorati et al. 2013 [34]	1 (11)	0.574 ± 0.006	0.34 ± 0.07	1 (7)	0.826 ± 0.014	0.69 ± 0.12
Migliorati et al. 2013 [34]	1 (11)	0.574 ± 0.006	0.34 ± 0.07	1 (7)	0.894 ± 0.006	0.58 ± 0.12
Sana et al. 2020 [41]	1 (12)	0.507 ± 0.010	0.138 ± 0.025	1 (7)	0.849 ± 0.024	0.142 ± 0.030
Sana et al. 2020 [41]	1 (12)	0.507 ± 0.010	0.138 ± 0.025	1 (11)	0.088 ± 0.049	0.181 ± 0.018
**Meta-analysis of in-vitro studies of peak load for pull-out strength—animal bone model**						
Radwan et al. 2017 [38]	10 (80)	0.70 ± 0.0	0.19411 ± 0.7392	10 (70)	0.83 ± 0.0	0.43699 ± 0.16779
Radwan et al. 2017 [38]	10 (80)	0.70 ± 0.0	0.19411 ± 0.7392	10 (70)	0.71 ± 0.0	0.25713 ± 0.07902
Radwan et al. 2017 [38]	10 (80)	0.70 ± 0.0	0.19411 ± 0.7392	10 (70)	0.89 ± 0.04	0.45062 ± 0.10022
Redžepagić-Vražalica et al. 2022 [44]	10 (80)	0.8 ± 0.006	0.16160 ± 0.566	20 (140)	0.890 ± 0.011	0.216.90 ± 0.568
Redžepagić-Vražalica et al. 2022 [44]	10 (80)	0.799 ± 0.006	0.16440 ± 0.5247	20 (140)	0.890 ± 0.011	0.216.90 ± 0.568
**Meta-analysis of in-vitro studies of peak load for pull-out strength—artificial bone model**						
**Author**	**Number of implants with reduced thread depth (number of threads-observations)**	**Dimension of thread depth [mm]**	**Pull-out strength [KN]**	**Number of implants with greater thread depth (number of threads-observations)**	**Dimension of thread depth [mm]**	**Pull-out strength [KN**
Migliorati et al. 2012 [33]	1 (6)	0.1735 ± 0.085	0.32 ± 0.05	1 (9)	0.01926 ± 0.0172	0.34 ± 0.07
Migliorati et al. 2012 [33]	1 (6)	0.1735 ± 0.085	0.32 ± 0.05	1 (10)	0.02757 ± 0.0093	0.43 ± 0.09
Migliorati et al. 2013 [34]	1 (11)	0.114 ± 0.01	0.34 ± 0.07	1 (7)	0.216 ± 0.013	0.58 ± 0.12
Migliorati et al. 2013 [34]	1 (11)	0.114 ± 0.01	0.34 ± 0.07	1 (7)	0.345 ± 0.029	0.69 ± 0.12
Sana et al. [41]	1 (11)	0.088 ± 0.019	0.181 ± 0.018	1 (12)	0.097 ± 0.027	0.138 ± 0.025
Sana et al. [41]	1 (11)	0.088 ± 0.019	0.181 ± 0.018	1 (7)	0.217 ± 0.046	0.142 ± 0.030
**Meta-analysis of in-vitro studies of peak load for pull-out strength—animal bone model**						
Radwan et al. 2017 [38]	10 (80)	0.22 ± 0.0	0.19411 ± 0.7392	10 (70)	0.22 ± 0.01	0.45062 ± 0.10022
Radwan et al. 2017 [38]	10 (80)	0.22 ± 0.0	0.19411 ± 0.7392	10 (70)	0.26 ± 0.0	0.43699 ± 0.16779
Radwan et al. 2017 [38]	10 (80)	0.22 ± 0.0	0.19411 ± 0.7392	10 (70)	0.33 ± 0.01	0.25713 ± 0.7902
Redžepagić-Vražalica et al. 2022 [44]	20 (140)	0.238 ± 0.017	0.216.90 ± 0.568	10 (80)	0.303 ± 0.005	0.16440 ± 0.5247
Redžepagić-Vražalica et al. 2022 [44]	20 (140)	0.238 ± 0.017	0.216.90 ± 0.568	10 (80)	0.272 ± 0.016	0.1616 ± 0.05663
**Meta-analysis of in-vitro studies of peak load for pull-out strength—artificial bone model**						
**Author**	**Number of implants with reduced thread shape factor (number of threads -observations)**	**Value of thread shape factor**	**Pull-out strength [KN]**	**Number of implants with greater thread shape factor (number of threads -observations)**	**Value of thread shape factor**	**Pull-out strength [KN]**
Migliorati et al. 2012 [33]	1 (6)	0.19 ± 0.01	0.32 ± 0.05	1 (9)	0.23 ± 0.02	0.34 ± 0.07
Migliorati et al. 2012 [33]	1 (6)	0.19 ± 0.01	0.32 ± 0.05	1 (10)	0.27 ± 0.01	0.43 ± 0.09
Migliorati et al. 2013 [34]	1 (11)	0.198 ± 0.018	0.34 ± 0.07	1 (7)	0.242 ± 0.015	0.58 ± 0.12
Migliorati et al. 2013 [34]	1 (11)	0.198 ± 0.018	0.34 ± 0.07	1 (7)	0.417 ± 0.038	0.69 ± 0.12
Sana et al. [41]	1 (12)	0.191 ± 0.05277	0.138 ± 0.025	1 (11)	0.20667 ± 0.04894	0.181 ± 0.018
Sana et al. [41]	1(12)	0.191 ± 0.05277	0.138 ± 0.025	1 (7)	0.25483 ± 0.04967	0.142 ± 0.030
**Meta-analysis of in-vitro studies of peak load for pull-out strength—animal bone model**						
Radwan et al. 2017 [38]	10 (70)	0.25 ± 0.02	0.45062 ± 0.10022	10 (80)	0.31 ± 0.01	0.19411 ± 0.7392
Radwan et al. 2017 [38]	10 (70)	0.25 ± 0.02	0.450.62 ± 0.10022	10 (70)	0.32 ± 0.01	0.43699 ± 0.16779
Radwan et al. 2017 [39]	10 (70)	0.25 ± 0.02	0.450.62 ± 0.10022	10 (70)	0.47 ± 0.01	0.25713 ± 0.7902
Redžepagić-Vražalica et al. 2022 [44]	20 (140)	0.270 ± 0.025	0.216.90 ± 0.568	10 (80)	0.340 ± 0.021	0.1616 ± 0.05663
Redžepagić-Vražalica et al. 2022 [44]	20 (140)	0.270 ± 0.025	0.216.90 ± 0.568	10 (80)	0.380 ± 0.006	0.16440 ± 0.05247

## Data Availability

Search data is available by corresponding author on reasonable request.

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
