# Peer review of "How Does Orthodontic Mini-Implant Thread Minidesign Influence the Stability?—Systematic Review with Meta-Analysis"

_jcm, 2022, doi:10.3390/jcm11185304_

Round 1
Reviewer 1 Report
Dear Authors,
Congratulations for your original and very interesting work.
The topic is really interesting and novel and can truly add new inspiration to current literature. The work is very well presented and structured.
However, I have some minor concerns that I would like to be addressed. You will find few comments/questions below.
- Discussion and Conclusions do not give a clear explanation on how the results can be a useful addition and contribution to literature and to the clinical practice. I suggesto to expand the Conclusion section specifically.
- Please check minor spelling errors.
- Figure 1 should be in high quality, please update it.
Once these issues will be solved - especially the Conclusion one
Author Response
Dear Reviewer thank You for for positive opinion please find the answers in intalics and changes in manuscript in red.
Dear Authors,
Congratulations for your original and very interesting work.
The topic is really interesting and novel and can truly add new inspiration to current literature. The work is very well presented and structured.
Thank You for Your positive opinion and kind comments.
However, I have some minor concerns that I would like to be addressed. You will find few comments/questions below.
- Discussion and Conclusions do not give a clear explanation on how the results can be a useful addition and contribution to literature and to the clinical practice. I suggest to to expand the Conclusion section specifically.
Conclusions have been revised, underlining clinical significance of the findings.
- Please check minor spelling errors.
Spelling has been cross-checked.
- Figure 1 should be in high quality, please update it.
The Figure 1 has been updated to high-quality image.
Once these issues will be solved - especially the Conclusion one
Reviewer 2 Report
Dear Authors
the paper is very interesting and can be considered for publication
however, some topics need to be addressed
1) What about implants of different collar design? Can it influence the stability? Please cite DOI10.3390/ma13051029
2) Which can be the role of such implants in patient underwent chemotherapy? Please cite PubMed ID26862696
3) What about digital workflow and orthodontic mini-implants? Please cite PubMed ID34425664
Author Response
Dear Reviewer,
thank You for for positive opinion please find the response in italics and changes in manuscript in red.
Dear Authors
the paper is very interesting and can be considered for publication
however, some topics need to be addressed
1) What about implants of different collar design? Can it influence the stability? Please cite DOI10.3390/ma13051029
2) Which can be the role of such implants in patient underwent chemotherapy? Please cite PubMed ID26862696
3) What about digital workflow and orthodontic mini-implants? Please cite PubMed ID34425664
Thank You for Your valuable comments and the citations proposed. They have been added to the discussion section.
Reviewer 3 Report
PICO and its components described starting in 69 are not very clear. What exactly is/are the outcome/s are you plan to use. Please correct your PICO and its components. Also, it is not clear if you mentioned clinical effectiveness as an outcome and then include in-vitro studies
78 please review this paragraph. It is not very clear, what exactly were the inclusion criteria you used. The results of the study: insertion torque?
Also, it is not mentioned if you use any language restriction.
Success as describe by?
106 You mentioned the wide range of studies as the reason to use MMAT. Consider reviewing the MMAT document since animal and in-vitro studies are not included. Excluding those studies in the appraisal.
129 Of the remaining 34 papers, only 1 could not have been retrieved. Correct sentence
130 studies head to be excluded, correct statement
134 as in 2 animal studies included studies, 1 in-vitro study and 4 finite element….. Unclear statement
172 Even if studies included in the review may seem possible to include in meta-analysis 172 they had to be excluded due to: presenting only correlations between studied factors, no 173 the raw numbers [33,39] or use of human cadaver heads [40]. Unclear statement.
181 why show a metanalysis with an I2 of 97.7?. Forest plot conclusions because of publication bias? Maybe the metanalysis was not indicated in the first place. Same for fig 5, fig 9, fig 12
183-190-195-201-207 correct the word “deminsion”
225 correct “deminsion infulence”
238 Correct infulence
279 how can you judge the quality of the evidence using a tool not appropriate to the research design?
288 studies included had to meet the requirement of including mini-288 mum three groups of subjects for comparison. Unclear statement, is this an inclusion criterion or you refer to the min number of studies to carry a metanalysis?
Discussion and conclusions require revision
Although the topic is important the methodology used requires major revision.
Because of the different study designs and the nature of the results consider using description of the findings instead of metanalysis.
Author Response
Dear Reviewer thank You for positive opinion. Please find the response in italics and changes in manuscript in red.
PICO and its components described starting in 69 are not very clear. What exactly is/are the outcome/s are you plan to use. Please correct your PICO and its components. Also, it is not clear if you mentioned clinical effectiveness as an outcome and then include in-vitro studies.
Thank You for You in-depth comment. In fact it was planned to find clinical studies, however, the review had to be modified. Studies on stability of mini-screws in terms of force needed to pull-out the screw out of the bone cannot be conducted on humans. It would absolutely unethical. We changed PICO and its components according to Your suggestions.
78 please review this paragraph. It is not very clear, what exactly were the inclusion criteria you used. The results of the study: insertion torque?
Also, it is not mentioned if you use any language restriction.
Success as describe by?
Thank You for Your kind note. We decided to remove the insertion torque as it is not directly associated with stability. No language restriction was applied. Main failure of the use of OMIs during orthodontic therapy is its loosening or falling out.
106 You mentioned the wide range of studies as the reason to use MMAT. Consider reviewing the MMAT document since animal and in-vitro studies are not included. Excluding those studies in the appraisal.
Animal and in-vitro studies are crucial in this field. Studies on stability of mini-screws in terms of force needed to pull-out the screw out of the bone cannot be conducted on humans. Thus such a studies o animal or artificial bone are the scientific method of choice. No studies have been concerning the survival rate of mini-screws in human patient regarding its minidesign. Due to this fact we excluded the MMAT as quality assessment tool and replaced it with QUIN, MQSSFE and SYRCLE. We hope the following correction will meet Your expectations.
129 Of the remaining 34 papers, only 1 could not have been retrieved. Correct sentence
Done.
130 studies head to be excluded, correct statement
Done.
134 as in 2 animal studies included studies, 1 in-vitro study and 4 finite element….. Unclear statement
172 Even if studies included in the review may seem possible to include in meta-analysis 172 they had to be excluded due to: presenting only correlations between studied factors, no 173 the raw numbers [33,39] or use of human cadaver heads [40]. Unclear statement.
Thank Your for Your note. The sentence has been corrected to “Even if studies included in the review may seem possible to include in meta-analysis they had to be excluded due to: presented only correlations between the examined factors, not specific values. [33,39] I the other study there was different research material used (human cadaver heads). [40]” I hope that now the corrected sentence provides information much more clearly.
181 why show a metanalysis with an I2 of 97.7?. Forest plot conclusions because of publication bias? Maybe the metanalysis was not indicated in the first place. Same for fig 5, fig 9, fig 12
Dear reviewer, thank you for your valuable suggestion. Indeed, this statement needs clarification. In my opinion such meta-analysis was absolutely indicated. The authors paid attention to very high heterogeneity in the discussion and were trying to find its sources. Additionally, in our manuscript there is no simple inference based on the results of meta-analysis with high heterogeneity. Thanks to the meta-analysis, the reader will be able to verify data present in the literature. Each one of included studies were based on same polyurethane foam block in case of artificial bone and in case of animal bone on similar species of animals in similar conditions on similar insertion depth. The studies included in each individual meta-analysis in this manuscript have several dozen citations, and their results are uncritically considered binding. We showed, especially in the case of TSF and tests carried out on the artificial bone model, that this should not be the case, which has been emphasized several times in the discussion. One should always remember, that the negative result is also a result.
183-190-195-201-207 correct the word “deminsion”
Done.
225 correct “deminsion infulence”
Done
238 Correct influence
Done.
279 how can you judge the quality of the evidence using a tool not appropriate to the research design?
As mentioned above.
288 studies included had to meet the requirement of including mini-288 mum three groups of subjects for comparison. Unclear statement, is this an inclusion criterion or you refer to the min number of studies to carry a metanalysis?
We corrected this sentence. This statement is referring to methodological requirements to possibly perform meta-analysis.
Discussion and conclusions require revision.
We changed this sections as kindly suggested.
Although the topic is important the methodology used requires major revision.
Because of the different study designs and the nature of the results consider using description of the findings instead of metanalysis.
Thank You for all the corrections proposed.
Reviewer 4 Report
The manuscript should be revised regarding grammar and spelling.
Why did the authors provided separately informations regarding
Authorship, year of 98 publication, type of each eligible study and its relevance regarding to the craniofacial mor- 99 phology of cleft palate only patients The central question of the manuscript seems quite rhetoric, giving all the physics orthodontic and implantology - bone related background. Conclusions are vague.
Author Response
Dear Reviewer thank You for positive opinion. Please find the answers in italics and changes in manuscript in red.
The manuscript should be revised regarding grammar and spelling.
Thank You, language editing was performed.
Why did the authors provided separately informations regarding
Authorship, year of 98 publication, type of each eligible study and its relevance regarding to the craniofacial mor- 99 phology of cleft palate only patients The central question of the manuscript seems quite rhetoric, giving all the physics orthodontic and implantology - bone related background.
Conclusions are vague.
Thank You for Your valuable suggestions. We corrected the typo and reformulated the conclusions. We hope that now the manuscript will meet Your expectations.
Reviewer 5 Report
1.Is the manuscript relevant and interesting?
The article is relevant and interesting.
2.How original is the topic?
The topic is current.
3.What does it add to the subject area compared with other published material?
The authors have collected and analyzed a huge quantity of data.
4. Is the paper well written?
Yes, the article is well written.
5. Is the text clear and easy to read?
Minor English editing is required.
6. Are the conclusions consistent with the evidence and arguments presented?
Yes, the conclusions consistent with the evidence and arguments presented.
7. Do they address the main question posed?
Yes, the Authors addressed the main question posed.
Other comments:
· English language: Minor English editing is required.
· Introduction: This section needs few improvements. For example, Authors may include a brief sentence at the beginning of this section regarding innovations in implant dentistry based on the following reference: <<Innovative materials and technologies to improve implants primary stabilityare an intense research topic in dentistry [https://doi.org/10.3390/jpm12010108]>>.
The other sections have been properly prepared.
Thanks for the opportunity to review this manuscript.
Author Response
Dear Reviewer, thank You for positive opinion. Please find the answers in italics and changes in manuscript in red.
1.Is the manuscript relevant and interesting?
The article is relevant and interesting.
2.How original is the topic?
The topic is current.
3.What does it add to the subject area compared with other published material?
The authors have collected and analyzed a huge quantity of data.
- Is the paper well written?
Yes, the article is well written.
- Is the text clear and easy to read?
Minor English editing is required.
- Are the conclusions consistent with the evidence and arguments presented?
Yes, the conclusions consistent with the evidence and arguments presented.
- Do they address the main question posed?
Yes, the Authors addressed the main question posed.
Other comments:
- English language: Minor English editing is required.
- Introduction: This section needs few improvements. For example, Authors may include a brief sentence at the beginning of this section regarding innovations in implant dentistry based on the following reference: <<Innovative materials and technologies to improve implants primary stabilityare an intense research topic in dentistry [https://doi.org/10.3390/jpm12010108]>>.
The other sections have been properly prepared.
Thanks for the opportunity to review this manuscript.
Thank You for appreciation of our manuscript. We introduced the suggested corrections. We hope that in current form it will meet your expectations.